# Ecological Niche and Interspecific Association of Plant Communities in Alpine Desertification Grasslands: A Case Study of Qinghai Lake Basin

**DOI:** 10.3390/plants11202724

**Published:** 2022-10-15

**Authors:** Ying Hu, Huichun Wang, Huiping Jia, Maodeji Pen, Nian Liu, Jingjing Wei, Biyao Zhou

**Affiliations:** 1College of Life Sciences, Qinghai Normal University, Xining 810008, China; 2Key Laboratory of Tibet Plateau Biodiversity Formation Mechanism and Comprehensive Utilization, Xining 810008, China; 3Key Laboratory of Medicinal Animal and Plant Resources on the Quinghai–Tibet Plateau, Xining 810008, China; 4College of Geographical Sciences, Qinghai Normal University, Xining 810008, China

**Keywords:** desertification gradient, niche breadth, niche overlap, interspecific linkages

## Abstract

The study of niche and interspecific relationships is one of the classical ecological theories. We set up four desertification gradients. The “Levins” and “Pianka” method were used to calculate the species’ niche breadth and niche overlap. Interspecies associations were analyzed by the ratio of variance (*VR*), Chi-square test, association coefficient (*AC*) and Ochiai index (*OI*). The results showed that in grasslands with different degrees of desertification, *Stellera chromosome* (3.90), *Thermopsis lanceolate* (3.52) and *Aster almanacs* (3.99) had larger niche widths, which were wide-area species of plant communities in the desertification area. The ecological niches of the same species in different habitats or different species in the same habitat were multi-dimensional. Niche differentiation measured by niche overlap can occur at any community succession stage. Niche width and niche overlap were not always consistent with environmental changes. Moreover, there was no linear relationship between them. The interspecific connection coefficient fluctuated greatly with the environment. The results can provide a reference for the study of plant community competition mechanism and desertification control in desertification land of the study area. We still do not know the mechanism of how the plants were preserved and how the retained plants adapted to the new environment during the desertification process. We can further study these questions in the next step.

## 1. Introduction

A niche refers to the ability of a population to maintain survival in a multi-dimensional space defined by a series of environmental axes, which not only explains the status (resource utilization) of the population or individual in the ecological space of the community [1], but also explains the coexistence and competition among species within the community [2]. Niche characteristics of community species mainly include niche breadth and niche overlap. Among them, niche breadth is a quantitative indicator of various resources that species can use, and niche overlap can take account of the competition of different biological populations for environmental resources. Species interconnection can express the relationship between different species in spatial distribution [3]. Niche research is important in understanding community structure and function, inter-species relationships within a community, biodiversity, community dynamics, succession [4] and population evolution. It is an area of focus for scholars [5,6,7,8].

Niche and interspecific association are widely used in ecological research. Wu et al. [9] found that with the extension of recovery time, the development of artificial grassland would lead to the increase in neutral interactions among plant species, and the proportion of positively and negatively correlated species in the community decreased, while the species-independent pairs increased significantly in the black soil flat artificial restoration grassland in the source of three rivers. With the extension of recovery time, the complexity of interspecies relationships (species network density) had a greater impact on community stability. Among the 20 dominant populations in the shrub layer of secondary natural Pinus tabulaeformis forest in Qin-ling Mountains, only 7.4% of the species pairs had significant positive association. There were significant correlations between species pairs, such as *Lespedeza bicolor* and *Elaeagnus pungens*, *Toxicodendron vernicifluum* and *Pinus tabuliformis*, *Euonymus alatus* and *Cerasus tomentosa*, but most species pairs had weak correlations. The ratio of positive correlation to negative correlation was less than 1, indicating that the interspecific association was weak. Species have certain independence in the dominant population [10]. In the karst hills of Guilin, Southwest China, the interspecific correlations of 22 major tree populations were significantly positive, indicating that the community was at a stable peak. Among the 231 species pairs, 108 species pairs were positively correlated, 115 species pairs were negatively correlated, and 8 species pairs were unrelated. The positive and negative linkage rate was 93.9%. Most species pairs did not show significant correlation, and the strong independence among species may be mainly due to the high heterogeneity of karst hilly habitats in Guilin, leading to the differentiation of ecological niche among species [11].

Desertification has caused severe ecological problems and it is a great challenge faced by arid and semi-arid regions around the world [12,13,14]. In 2014, the area of desertification land in China reached 1.72 × 10^6^ km^2^, mainly aeolian desertification [15]. Land desertification will lead to the decline of productivity, loss of biodiversity and weakening of ecological functions, seriously restricting the development of human society [16]. The Qinghai–Tibet Plateau is also facing the acute problem of desertification. According to the research of Cuo et al. [17], there are four types of desertification in the Quinghai–Tibet Plateau, among which the area of moderate desertification is of largest importance (2.53 × 10^5^ km^2^), followed by light desertification (1.06 × 10^5^ km^2^) and extreme heavy desertification (3.00 × 10^4^ km^2^), with the lowest coverage area being severe desertification (1.76 × 10^4^ km^2^).

The Qinghai Lake Basin is located in the northeastern part of the Quinghai–Tibet Plateau [18]. It plays an important ecological barrier function in preventing the further westward migration of desertification [19], maintaining the ecological security of the entire Quinghai–Tibet Plateau [20], ensuring the normal continuation of life for human beings on the plateau [21,22] and protecting its natural habitat for plant and animal species [23]. Research on sandy land in the Qinghai Lake Basin mainly focuses on the use of remote sensing technology to monitor the dynamic changes of sandy land [24], the classification of desertified land [25], the evolution of sandy land [14] and the study of individual vegetation (such as the water transport mechanism of Achnatherum splendors [26]), etc. However, there are few reports on the ecological niche and interspecific association of desertification plants in the Qinghai Lake basin. We hypothesized that “although the climatic and topographic factors of the Quinghai–Tibet Plateau are unique, the overall trend of vegetation competitive succession in the desertified process is basically the same”. Essentially, this study took 17 species of plants in the sandy area of the Qinghai Lake Basin as the research object, and based on species importance values, analyzed species niche characteristics, overall community connectivity and interspecific relationships under different desertification stages, trying to explain the response of major plant species to different desertification stages from the perspective of ecological niche and interspecies relationship, and understand the influence of the desertification process on the structure and function of the alpine grassland ecosystem and its plant community, to provide a theoretical basis for the prevention and control of grassland desertificattion.

## 2. Results

### 2.1. Niche Width

Under different desertification stages, there were specific differences in niche width of the 17 main species in the study area (Table 1). Among the species with the largest niche breadth in Levins, the non-desertified grasslands were *Stellera chamaejasme* (3.90), *Kobresia humilis* (3.83) and *Thermopsis lanceolata* (3.52), the light desertification grasslands were *Aster Altaicus* (3.99), *Dracocephalum heterophyllum* (3.83) and *Thermopsis lanceolata* (3.52), the moderately desertified grasslands were *Poa pratensis* (2.97), Lancea tibetica (2.81) and Pedicularis kansuensis (2.76), and heavy desertification grasslands were Stellera chamaejasme (3.72), *Thermopsis lanceolatea* (2.73) and Aster Altaicus (1.99). With the intensification of desertification, the species composition of plant communities will change.

### 2.2. Niche Overlap

Under different desertification stages, there were differences in the niche overlap of 17 major species. When the overlap value was greater than 0.83, there were 14 pairs (10.37%) of not desertification, 12 pairs (8.89%) of light desertification, 17 pairs (12.59%) of moderate desertification, and 20 pairs (14.81%) of heavy desertification. When the overlap value was between 0.67 and 0.83, there were 18 pairs (13.33%) not desertification, 12 pairs (8.89%) with light desertification, 14 pairs (10.37%) with moderate desertification and 5 pairs (3.70%) with heavy desertification. When the overlap value was between 0.50 and 0.67, there were 11 pairs (8.15%) not desertification, 23 pairs (17.04%) with light desertification, 13 pairs (9.63%) with moderate desertification and 15 pairs (11.11%) with heavy desertification. When the overlap value was between 0.33 and 0.50, there were 23 pairs (17.04%) of not desertification, 17 pairs (12.59%) of light desertification, 18 pairs (13.33%) of moderate desertification and 16 pairs (11.85%) of heavy desertification. When the overlap value was between 0.17 and 0.33, there were 19 pairs (14.07%) not desertification, 10 pairs (7.41%) with light desertification, 20 pairs (14.81%) with moderate desertification and 8 pairs (5.93%) with heavy desertification. When the overlap value was less than 0.17, there were 34 pairs (25.19%) of not desertification grassland, 46 pairs (34.07%) of moderate desertification grassland, 33 pairs (24.44%) of moderate desertification grassland and 54 pairs (40.00%) of heavy desertification grassland. The logarithmic range of each ecological overlap level distribution in the not, lightly, moderately and heavy desertified grasslands were 23, 36, 20 and 49, respectively, indicating that in terms of distribution dispersion, the heavy desertification grasslands were the largest and the moderate desertification grasslands were the smallest. In general, there were significant differences in the resource utilization degree of 17 plant species in each desertification gradient. The degree of interspecific competition between not desertification and moderate desertification grassland was similar and larger, while the heavy desertification grassland had the smallest (Figure 1).

### 2.3. Analysis of Overall Association between Multiple Species

#### 2.3.1. Overall Connectedness

*VR* value of each type of grassland was greater than 1, indicating that the overall connectivity of the 17 species in this area showed positive connectivity. The calculation results of the statistic *W* showed that the *W* value of the light and moderately desertified grassland did not fall within the chisquare critical value range, indicating that the overall connectivity of the main species in these two grassland types was significantly correlated. Entering the chi-square critical value indicates that the overall connectivity of the principal species is not significant (Table 2).

#### 2.3.2. Chi-Square Statistic Test

Among the 135 species pairs, there were 17 significant positive-linked species pairs (*p* < 0.05) in the not desertification grasslands, accounting for 12.59%, without negatively linked species pairs; the other types of desertification grassland types were all positively linked species pairs, without significant positive association (Figure 2).

#### 2.3.3. Association Coefficient

Among the grassland *AC* connection coefficients in each desertification stage, the number of not desertification, slight, moderate and heavy desertification grasslands with *AC* ≥ 0.67 was 31, 48, 47 and 41 pairs; the number of 0.33 ≤ *AC* < 0.67 was 11, 7, 12 and 13 pairs; 0.00 ≤ *AC*< 0.33 was 46, 14, 34 and 23 pairs; −0.33 ≤ *AC* < 0.00 was 9, 10, 6 and 4 pairs; −0.67 ≤ *AC* < −0.33, except for 2 pairs of not desertification grassland, the other grassland types were not distributed; the number of *AC* < −0.67 was 36, 56, 36 and 54 pairs. In each type of grassland, the logarithms of positive linkages accounted for 65.19%, 51.11%, 68.89% and 57.04%, respectively, and the logarithms of negative linkages accounted for 34.81%, 48.89%, 31.11% and 42.96%. As the degree increased, the number of positive-linked plant pairs decreased and the number of negative-linked plant pairs increased (Figure 3).

#### 2.3.4. OI Semi-Exponential Matrix

Among the four desertified grasslands, 135 plant species had specific differences in the degree of interspecific association. The number of not desertification, slight, moderate and heavy desertification grasslands with 0.83 ≤ *OI* was 17, 7, 11 and 15 pairs; the 0.67 ≤ *OI* < 0.83 was 18, 31, 27 and 42 pairs; 0.50 ≤ *OI* < 0.67 was 47, 41, 41 and 20 pairs; 0.33 ≤ *OI* < 0.50 was 23, 14, 34 and 4 pairs; 0.17 ≤ *OI* < 0.33 was 2 for the not desertification grassland, and the other grassland types were no distribution; *OI* < 0.17 was 28, 41, 22 and 53 pairs. In the statistics of 0.50 ≤ *OI*, there were 82 pairs of not desertification grasslands, accounting for 60.74%; 79 pairs of light dissatisfied grasslands, accounting for 58.52%; and 79 pairs of moderately desertified grasslands, accounting for 58.52%, and 0.50 pairs of heavy dissatisfied grasslands was 78 pairs, accounting for 57.78% (Figure 4).

#### 2.3.5. Correlation Analysis of Different Desertification Plots

The cluster analysis of the important values of the main species in the sandy area of the Qinghai Lake Basin, according to the degree of desertification, can be divided into four types: not, light, moderate and heavy desertification. One class was composed of *Thermopsis lanceolata* and *Stellera chamaejasme*, the second class was *Taraxacum scariosum*, *Aconitum gymnandrum*, *Silene jenisseensis*, *Achnatherum splendens*, *Lancea tibetica* and *Pedicularis kansuensis*, the third class was *Medicago sativa*, *Gentiana macrophylla*, *Elymus nutans*, *Poa pratensis*, *Allium przewalskianum*, *Oxytropis kansuensis* and *Kobresia humilis*, and the fourth group was *Aster altaicus* and *Dracocephalum heterophyllum* (Figure 5).

## 3. Discussion

It was of great significance to explore the plant species and community succession mechanism in the process of grassland desertification on the Quinghai–Tibet Plateau [27,28]. The results showed that the desertification process had a specific effect on the plant niche width, and the same environmental conditions had different effects on different plant species. For example, with the increase in desertification degree, the niche width of the *Kobresia humilis* decreased gradually, and the niche width of the *Dracocephalum heterophyllum* fell after rising, which was consistent with Edouard Ilunga wa Ilunga et al.’s research on the niche of the small-scale plant community on the Katanga copper outcrop in Congo (DRC) [29]. Niche width is a measure of a species’ ability to use resources in a certain area. The niche width of the same species varies under different environmental conditions. For example, this study showed that *Elymus nutans* was larger in not desertification and moderate desertification grassland, and smaller in severe desertification grassland, which was similar to Ulises Rodriguez-robles et al. [30]. They found that oak and pine trees in semi-arid tropical forests in central Mexico had consistent multi-dimensional niche results in different seasons.

Competition among plant communities has become a consensus of most scientists [31,32,33,34]. Classical niche theory suggests that each species can survive under limited conditions, and that a large overlap of limiting factors prevents a species from gaining a foothold in a community [35]. This study found an ecological niche overlap of major plant species in the sandy area of Qinghai Lake Basin. The number of coupling plant pairs in the not desertification and moderately desertified grassland was larger than in the heavy desertified grassland with the overlap value ≥0.33, and the competition was more intense. This may be due to the fact that the desertified grassland ecosystem is in a state of balance, and plant species need to increase interspecific competition in the limited living space to obtain energy for survival and reproduction, while the ecological balance of mildly desertified grassland is broken, and the plant community is in an unbalanced state. When the environment transitions to moderate degradation, plant species redistribute living resources and increase competition. Subsequently, the environment continues to deteriorate, the number of plant species decreases and niches diverge after adaptation, the overlap value of niches decreases and competition decreases. Plant species tend to coexist. This is consistent with the results of Silvertown et al. [36], who found that the expected changes of niche overlap of meadow plants on hydrological gradients are different, and niche separation can occur at all phylogenetic levels. This change is similar to the “moderate disturbance hypothesis” of the grassland ecosystem on the Quinghai–Tibet Plateau [37], but it needs to be further verified.

Overall, interspecific associations reflect the stability of plant communities and are essential parameters of vegetation succession [38]. They result from interactions between species, interactions in the food chain and similar responses and adaptations to environmental forces [39,40,41]. The results of this study show that there was a significant positive correlation between non-desertification and moderate desertification grassland. At the same time, there was no significant positive correlation between mild and heavy desertification grassland. With the increase in desertification degree, the number of positive coupling plant pairs decreased while the number of negative coupling plant pairs increased. This indicated that the undesertification grassland plant community was in a relatively stable state, while the severely desertified grassland plant community was in an unstable state. In general, the stability of community structure and species composition increased with community succession [42].

## 4. Materials and Methods

### 4.1. Overview of the Study Area

The Qinghai Lake basin is between 36°15′~38°20′ N, 97°50′~101°20′ E, 3194~5174 m above sea level, and has an area of 2.97 × 10^4^ km^2^. This intersection of the arid northwest, alpine southwest and monsoon east in China is a climate-sensitive and ecologically fragile region of the world [43] (Figure 6). The climate is a typical semi-arid alpine plateau climate, with an average annual temperature of −1.1~4.0 °C, rainfall of 300~450 mm, concentrated from June to September, and annual evaporation of about 930 mm, which has the function of adjusting the local microclimate [44]. The soil types are predominantly sandy soil, meadow soil and saline–alkali soil. There are many dunes on the east bank of Qinghai Lake of different height. The highest dunes are higher than 100 m. The types of dunes are mainly crescent-shaped, crescent-shaped chain and dune network types, and a few are pyramid-shaped. These dunes generally move from northwest to southeast, in line with the extension of the sand belt [45,46]. The main vegetation types are meadow (1.66 × 10^4^ km^2^) and grassland (4.60 × 10^3^ km^2^).

### 4.2. Meths

#### 4.2.1. Sample Set

According to the national standard of the People’s Republic of China “Parameters for degradation, sandification and salification of rangelands” (GB 19377-2003) [47], 4 types of grassland desertification (not desertification, slight, moderate and heavy desertification) were set up. From June to September 2021, a completely random block design was used in each sample plot in the sandy area around the lake in Qinghai Lake. Each plot was set with 5 quadrats, with 20 samples. The size of each quadrat was 1 m × 1 m, and the grassland plant species, coverage, height, frequency, and abundance were investigated in the quadrat. The plants used in the analysis were required to be present in each desertification type [48] at the same time and the importance value was greater than 0.5%.

#### 4.2.2. Computational Formula


(1)Niche breadth and niche overlap


Niche width refers to the sum of all kinds of resources that can be used by organisms; that is, an indicator of the diversity of biological utilization resources. The wider the niche of a species, the less specialized it is; that is, the more likely it is to be a generalized species. When the reverse is the case, it tends to be a specialized species. Niche overlap refers to the phenomenon that two or more species with similar ecological niches share or compete for common resources when they live in the same space. Two species with overlapping niches cannot coexist for long periods of time unless space and resources are abundant due to the principle of competitive exclusion. There is usually a limit on resources, so competition between species with overlapping niches will always lead to less overlap.
Important value = (relative density + relative cover + relative frequency)/3(1)

The “Levins” method was used to calculate the niche breadth, and the “Pianka” method was used to calculate the niche overlap [38].
(2)$$B_{i}=\frac{1}{\sum_{j=1}^{r}{\left(\ln P_{ij}\right)}^{2}}$$
(3)$$O_{ik}=\frac{\sum_{j=1}^{r}P_{ij}P_{kj}}{\sqrt{\sum_{j=1}^{r}P_{ij}{}^{2}\sum_{j=1}^{r}P_{kj}{}^{2}}}$$

In Formulas (2) and (3), *B_i_* is the niche breadth of species *i*, *O_ik_* is the niche overlap value of species *i* and species *k*, and *P_kj_* are the important values of species *i* and species *k* on treatment *j*, respectively.


(2)Overall connectedness




(4)
$$\delta_{T}^{2}=\overset{s}{\underset{i=1}{\sum }}P_{i}\left(1-P_{i}\right)$$


(5)
$$S_{T}^{2}=\left(\frac{1}{N}\right)\overset{N}{\underset{j=1}{\sum }}{\left(T_{j}-t\right)}^{2}$$


(6)
$$P_{i}=\frac{n_{i}}{N}$$


(7)
$$VR=\frac{s_{T}^{2}}{\delta_{T}^{2}}$$


(8)
W=VR×N



In Formula (5), *S* is the total number of species in the survey plot; *N* is the total number of squares; *n_i_* is the number of squares in which species i appears; *T_j_* is the total number of species that appear in square *j*; *t* is the average of species in all squares numbered. In Formula (7), *VR* > 1 means a positive association between species, and *VR* < 1 means a net negative association between species. In Formula (8), the statistic *W* = *VR* × *N* is used to test the significance of the *VR* value deviating from 1. If the species are not significantly associated, the probability of *W* falling within the limit is given by the following *χ^2^*_0.95_(*N*) < *W* < χ^2^_0.05_(*N*) [49].


(3)Interspecific association


The measured species were arranged in a 2 × 2 contingency table χ^2^ by presence or absence data, and correlation calculations and interspecies linkage degree analysis were performed.


(1)Statistical magnitude


Interspecies association is one of the important quantitative and structural characteristics of plant communities, which is the basis for the formation and evolution of plant community structure.

The 2 × 2 contingency table statistic can test interspecific association and significance. Yates’ continuous correction factor was used to correct statistical errors caused by sampling discontinuities. The formula is as follows [50]:(9)$$\chi^{2}=\frac{N{\left(\left|ab-bc\right|-0.5N\right)}^{2}}{\left(a+b\right)+\left(c+d\right)\left(a+c\right)\left(b+d\right)}$$

In the Formula (9), *a* is the quadrat where both species appear, *b* and *c* are the quadrat numbers where only species 1 and species 2 appear, respectively, and *d* is the quadrat number where neither species appear. When *χ*^2^ < 3.814, the interspecific association is not significant (*p* > 0.05), and the interspecific association is considered basically independent; when *χ*^2^ > 3.814, the interspecific association is considered significant (*p* < 0.05). When *ad* > *bc*, there is a positive association between species. On the contrary, when *ad* < *bc*, there is a negative association between species.


(2)Connection strength




(10)
$$AC=\frac{ad-bc}{\left(a+b\right)\left(b+d\right)}\;\;(\mathrm{ad}\;\geq \;\mathrm{bc})\;$$


(11)
$$AC=\frac{ad-bc}{\left(a+b\right)\left(a+c\right)}\;(\mathrm{bc}\;>\;\mathrm{ad},\;d\;\geq \;a)$$


(12)
$$AC=\frac{ad-bc}{\left(a+b\right)\left(d+c\right)}\;(\mathrm{bc}\;>\;\mathrm{ad},\;d\;>\;a)$$


(13)
$$OI=\frac{a}{\sqrt{\left(a+b\right)\left(a+c\right)}}$$



In the Formulas (10)–(13), *a* is the quadrat where both species appear, *b* and *c* are the numbers of quadrat where only species 1 and 2 appear, respectively, and *d* is the number of quadrat where neither species appears [51].

Microsoft Excel 2019 software was used for data sorting and statistics. R language (R i386 4.1.3, data package using “spaa”) completed the calculation of niche width, niche overlap and overall connectedness. ARC GIS (ARCMAP 10.8) drew a schematic diagram of sample points.

## 5. Conclusions

With the intensification of desertification, the ecological niche of plateau grassland has been gradually differentiated, competition has been gradually reduced, and a regional coexistence pattern of plant community has been observed. This study simulated the degradation–succession process of the main plant species niche and the connections created by the changed situation for the study of desertification in the succession of plant community characteristics. This provides a new quantitative method, at the same time, to explore the desert plant competition mechanism in its natural environment. The severe desertification of the Quinghai–Tibet plateau, and alpine grassland recovery, are very important issues. However, we still do not understand the mechanism by which the plants were preserved and how the retained plants adapted to the new environment during the desertification process. We can further study these questions in the next step.

## Figures and Tables

**Figure 1 plants-11-02724-f001:**
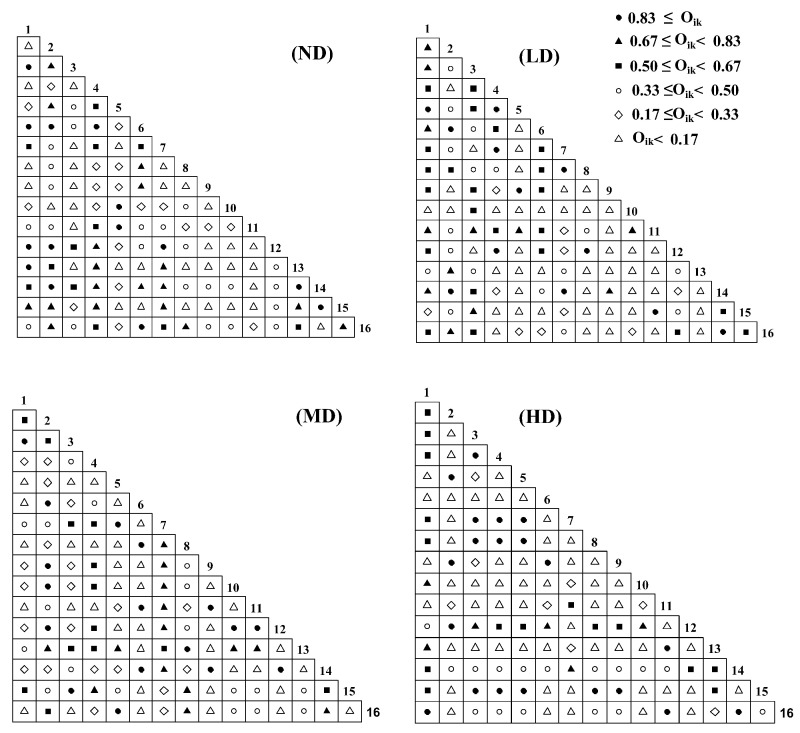
Niche overlap values of major species at different desertification stages in the Qinghai Lake Basin. Values 1–16 outside the table represent different plant species. ND stands for not desertification, LD for light desertification, MD for moderate desertification, HD for heavy desertification. See Table 1 for the meanings of species numbers.

**Figure 2 plants-11-02724-f002:**
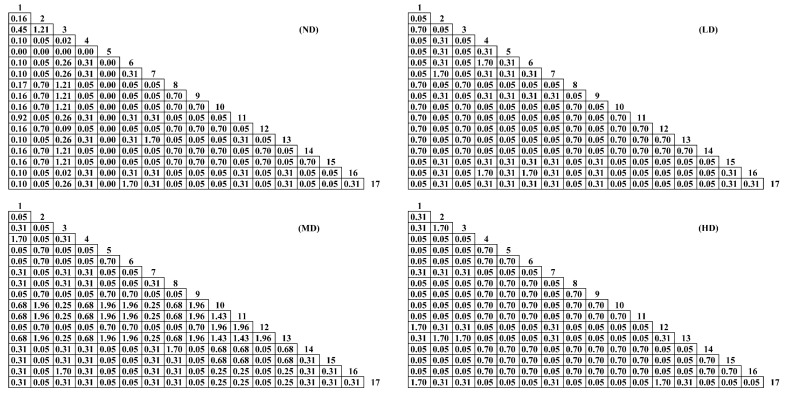
Chi-square test of *p*-value matrix of main species in different desertification stages in Qinghai Lake basin. This graph represents the significance level of the chi-square test for different plant species. ND stands for not desertification, LD for light desertification, MD for moderate desertification, HD for heavy desertification. See Table 1 for the meanings of species numbers.

**Figure 3 plants-11-02724-f003:**
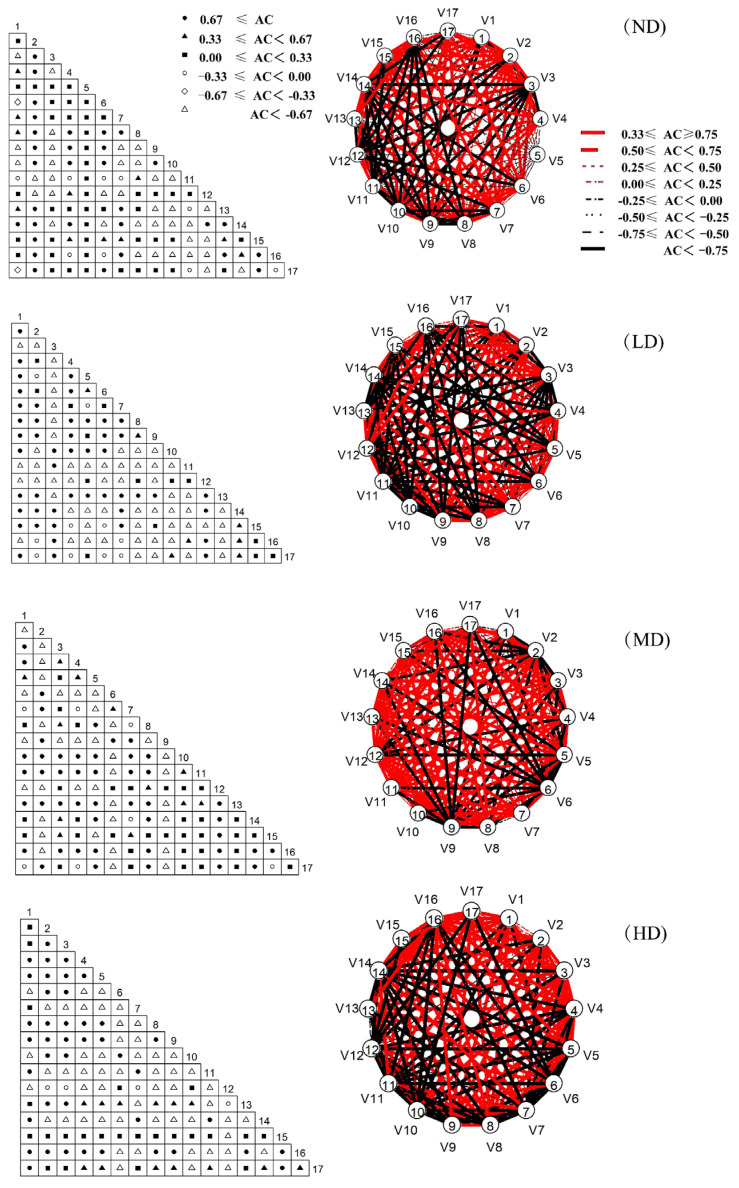
*AC* (association coefficient) linkage coefficients of major species in different desertification stages in the Qinghai Lake Basin. The numbers 1–17 represent different plant species, and the circular graph on the right represents the government association of different species. ND stands for not desertification, LD for light desertification, MD for moderate desertification, HD for heavy desertification. See Table 1 for the meanings of species numbers.

**Figure 4 plants-11-02724-f004:**
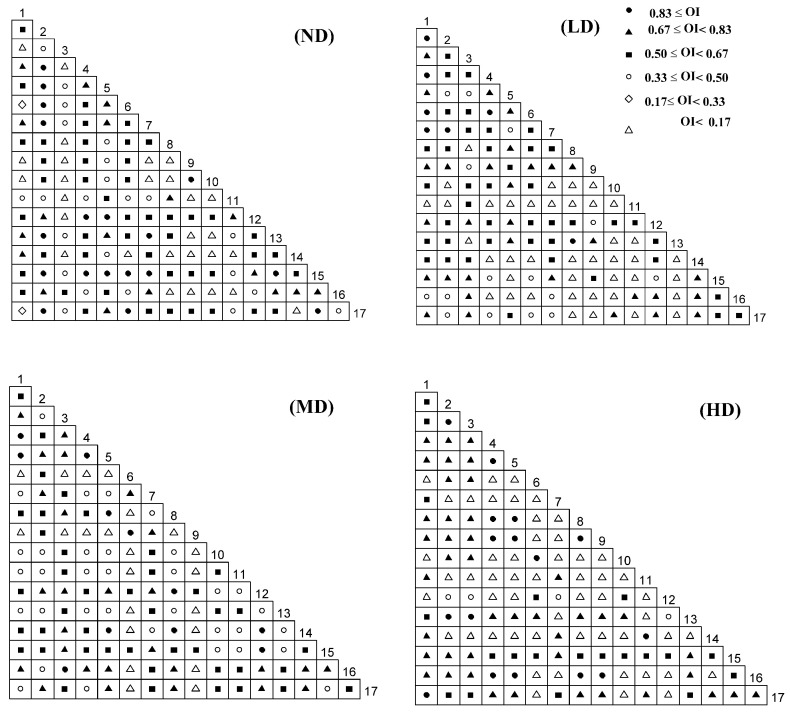
*OI* (Ochiai index) semi-exponential matrix of major species in different desertification stages in the Qinghai Lake Basin. ND stands for not desertification, LD for light desertification, MD for moderate desertification, HD for heavy desertification. See Table 1 for the meanings of species numbers.

**Figure 5 plants-11-02724-f005:**
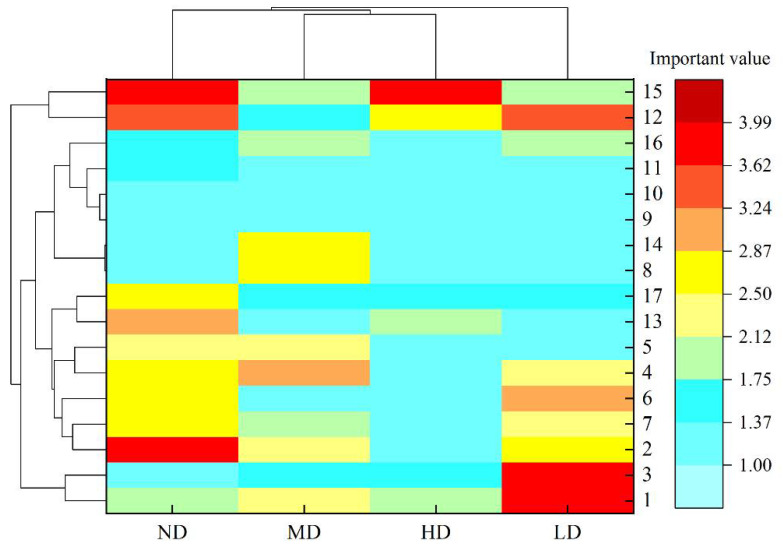
Correlation analysis of important values of main plants in the sandy area of Qinghai Lake Basin. ND stands for not desertification, LD for light desertification, MD for moderate desertification, HD for heavy desertification. See Table 1 for the meanings of species numbers.

**Figure 6 plants-11-02724-f006:**
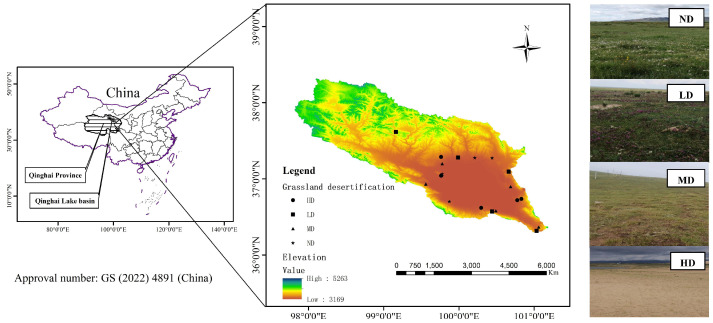
Location diagram and sample setting of Qinghai Lake Basin (China). The figure on the left shows the geographical position of Qinghai Province in China and the Qinghai Lake Basin in Qinghai Province. Meanwhile, our survey plots are also marked in the geographical map of Qinghai Lake basin. On the right, from top to bottom, are the habitat photos of not (ND), light (LD), moderate (MD) and heavy (HD) desertification plots. The dataset of Qinghai Lake Basin is provided by National Cryosphere Desert Data Center. (http://www.ncdc.ac.cn (accessed on 17 August 2022)).

**Table 1 plants-11-02724-t001:** Levins niche widths of main plant species in different desertification stages. This table shows the quantified ecological widths of 17 major species in different desertification stages.

Number	Species	Desertification Gradient			
		Not Desertification	Light Desertification	Moderate Desertification	Heavy Desertification
1	*Aster altaicus*	1.91	3.99	2.37	1.99
2	*Kobresia humilis*	3.88	2.66	2.26	1.24
3	*Dracocephalum heterophyllum *	1.00	3.83	1.56	1.40
4	*Poa pratensis*	2.67	2.19	2.97	1.00
5	*Elymus nutans*	2.41	1.31	2.47	1.00
6	*Allium przewalskianum*	2.73	2.99	1.00	1.00
7	*Oxytropis kansuensis*	2.81	2.49	2.00	1.35
8	*Pedicularis kansuensis*	1.00	1.00	2.76	1.00
9	*Achnatherum splendens*	1.00	1.22	1.00	1.00
10	*Silene jenisseensis*	1.00	1.00	1.00	1.00
11	*Aconitum gymnandrum*	1.63	1.00	1.00	1.00
12	*Thermopsis lanceolata*	3.35	3.52	1.67	2.73
13	*Gentiana macrophylla*	2.92	1.00	1.00	1.99
14	*Lancea tibetica*	1.00	1.00	2.81	1.00
15	*Stellera chamaejasme*	3.90	2.00	1.89	3.72
16	*Taraxacum scariosum*	1.41	1.92	1.78	1.00
17	*Medicago sativa*	2.81	1.57	1.72	1.65

**Table 2 plants-11-02724-t002:** Overall correlation of major species in different desertification stages in the Qinghai Lake Basin. The table gives the main calculation values of the overall connectedness calculation process and the significance test of the probability.

Desertification Gradient	*δ_r_^2^*	*S_r_^2^*	*VR*	*W*	Chi Squared Critical [N_0.95_, N_0.05_]	*p*	Result
ND	3.15	8.56	2.72	13.59	[3.00, 10.00]	*p* < 0.05	Significant positive correlation
LD	3.44	4.16	1.21	6.05	[6.00, 10.00]	*p* > 0.05	No significant positive correlation
MD	3.25	11.29	3.47	17.36	[4.00, 13.64]	*p* < 0.05	Significant positive correlation
HD	3.28	11.84	3.61	18.05	[3.00, 20.00]	*p* > 0.05	No significant positive correlation

## Data Availability

The datasets used during the current study are available on reasonable request.

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
