# Peer review of "Ecological Niche and Interspecific Association of Plant Communities in Alpine Desertification Grasslands: A Case Study of Qinghai Lake Basin"

_plants, 2022, doi:10.3390/plants11202724_

Round 1

Reviewer 1 Report

The authors analyzed the ecological niche characteristics of species, overall community connectivity and interspecific relationships under different stages of desertification using 17 plant species in the sandy areas of the Qinghai Lake Basin. The results of the study have some significance for understanding the effects of desertification process on alpine grassland ecosystem and its plant community structure and function, as well as preventing and controlling grassland desertification. However, there are still many shortcomings suggested for improvement.

1. The current research progress on niche and interspecific relationships are not described in Introduction, and it is suggested to supplement.

2. Table 1 is suggested to be changed to a spatial schematic diagram of sample sites for more intuitive understanding by readers.

3. In Table 1, what does' ND ', 'HD', 'MD' and 'LD' represent respectively? Please add a note.

4. L51: 'the area of diversified land' , diversified?

5. L134 and L142, please check the formula number.

6. L147: No sampling site schematic in the main text.

7. L147: 'mild', L155: 'the slight desertification grasslands', Table 3: 'light desertification'. The expression should be uniform, and the full text should be checked.

8. L153: 'the not desertification grasslands', L155: 'the slight desertification grasslands', L156: 'the moderately esterified grasslands', L157: 'the severely degraded grasslands'. Please use the correct expression and be consistent with the 4 types in Table 3 for easy reading.

9. L162: Not Table 2?

10. Why is there only 1-16 in Figure 1.

11. Please check the expression at L183-L185 'indicating ..... smallest'.

12. L207: 17?

13. In '3.3.2. Chi-square statistic test', why is the positive and negative correlation illustrated by the p-value?

14. Why is the legend different in Figure 3?

15. L231: esterified? Please check the full text.

16. L247: three types?

17. What is the basis for the classification of Oik, AC and OI indicators?

18. The format of references needs to be further standardized, for example, many references are missing page numbers.

19. English needs to be further improved to increase readability.

Author Response

Question 1: The current research progress on niche and interspecific relationships are not described in Introduction, and it is suggested to supplement.

Reply 1: We have included relevant research developments. L49-L71

Question 2: Table 1 is suggested to be changed to a spatial schematic diagram of sample sites for more intuitive understanding by readers.

Reply 2: We are changed to a spatial schematic diagram of sample sites. L296

Question 3: In Table 1, what does' ND ', 'HD', 'MD' and 'LD' represent respectively? Please add a note.

Reply 3: We turn Table 1 into Figure 1 and add notes of 'ND', 'HD', 'MD' and 'LD' represent. L296

Question 4: L51: 'the area of diversified land' , diversified?

Reply 4: Modified to the area of desertification land.  L109, L126, L130, L157, L185 ,L203,

Question 5: L134 and L142, please check the formula number.

Reply 5: We checked the formula number and corrected the error. L325-L360

Question 6: L147: No sampling site schematic in the main text.

Reply 6: We added a sampling site map to the geographic map (Figure 1). L293

Question 7: L147: 'mild', L155: 'the slight desertification grasslands', Table 3: 'light desertification'. The expression should be uniform, and the full text should be checked.

Reply 7: We have unified the full text on the degree of desertification terms, Not desertification, Light desertification, Moderate desertification and Heavy desertification.

Question 8: L153: 'the not desertification grasslands', L155: 'the slight desertification grasslands', L156: 'the moderately esterified grasslands', L157: 'the severely degraded grasslands'. Please use the correct expression and be consistent with the 4 types in Table 3 for easy reading.

Reply 8 : We checked the text and agreed on these terms. Not desertification, Light desertification, Moderate desertification and Heavy desertification.

Question 9: L162: Not Table 2?

Reply 9 : We have merged the contents of Table 2 into Table 3. L123

Question 10: Why is there only 1-16 in Figure 1.

Reply 10 : Figure 1 shows the situation of niche overlap values. Niche overlap refers to the phenomenon that two or more species with similar ecological niches share or compete for common resources when they live in the same space. The niche overlap value is used to quantify the number of species that share or compete for common resources when two or more species with similar niches live in the same space. Because of the pairwise comparison, they are always 1 less than the number of species.

Question 11: Please check the expression at L183-L185 'indicating ..... smallest'.

Reply 11 : Let's rewrite this sentence to “indicating that in terms of distribution dispersion, the heavy desertification grasslands were the largest and the moderate desertification grasslands were the smallest”. L144-L145

Question 12: L207: 17?

Reply 12 : We change it to 17 plant species. L170

Question 13: In '3.3.2. Chi-square statistic test', why is the positive and negative correlation illustrated by the p-value?

Reply 13 : Positive association is the same or similar to the adaptation or response of plants to the habitat, while negative association is not the same or similar to the adaptation or response of plants to the habitat.

Question 14: Why is the legend different in Figure 3?

Reply 14 : We changed the legend of Figure 3. L192

Question 15: L231: esterified? Please check the full text.

Reply 15 : We modify esterified in appropriate locations of the full text to desertified. L161, L199, L251, L252, 253,

Question 16: L247: three types?

Reply 16 : We will correct the writing errors to four types.

Question 17: What is the basis for the classification of Oik, AC and OI indicators?

Reply 17: They were divided into 6 levels from largest to smallest, representing significant positive correlation, insignificant positive correlation, insignificant negative correlation, insignificant negative correlation and significant negative correlation.

Question 18: The format of references needs to be further standardized, for example, many references are missing page numbers.

Reply 18: We downloaded the "endnote" bibliographic reference format for the journal from the official website, added the DOI of the cited literature, and corrected the journal abbreviation.

Question 19: English needs to be further improved to increase readability.

Reply 19: We submitted the paper to the MDPI English Grammar Review and Revision Project and completed the grammar revision.

Reviewer 2 Report

Dear authors,

I have reviewed the manuscript titled "Study on the ecological niche and interspecific association of plant communities in alpine desertification grasslands on the Qinghai-Tibet Plateau: A case study of Qinghai Lake Basin". The topic is interesting, the language is acceptable, and the article is fit for the journal ‘Plants’ but can't accept in its present form. Because, the title, abstract, and results need improvement. Moreover, the legends of Figures are not clear and satisfactory so need to re-write them. I believe that the article can be accepted for publication in the ‘Plants’ journal with minor revision. Find my comments enclosed in the manuscript.

Author Response

Question 1: L1 There should be space before the citation/reference number, so please check the whole manuscript and address this issue.

Reply 1: We checked and corrected the full text for all such issues.

Question 2: Line 2-4: Please modify the title as ' Ecological niche and Interspecific association of Plant communities in Alpine Desertification Grasslands: A case study of Qinghai Lake Basin'.

Reply 2 : We have modified the title as ' Ecological niche and Interspecific association of Plant communities in Alpine Desertification Grasslands: A case study of Qinghai Lake Basin'.

Question 3: Line 15-34: The abstract should be a total of about 200 words maximum as per journal policy. So, please reduce it accordingly.

Reply 3 : We have revised the abstract and it is now less than 200 words. L16-L29

Question 4: Line 16: Please delete 'in this study'.

Reply 4 : We delete 'in this study' of line 16. L17

Question 5: Line 33: Please remove this keyword 'Qinghai Lake basin;' because you already used this word in your title.

Reply 5: We remove the keyword 'Qinghai Lake basin;'. L30

Question 6: Line 37: Please add 'A' before niche refers.....

Reply 6: We add 'A' before niche refers...... L33

Question 7: Line 41: Please delete 'is one of the basic theories of mechanism'

Reply 7:We delete 'is one of the basic theories of mechanism'. L37

Question 8: Line 51: Please consider add/cite these references here:

https://doi.org/10.1016/j.jplph.2022.153671

Reply 8: We added this reference. L 425 [12]

Question 9: Line 55: Please delete 'with',

Please replace 'Depending  on  the  research  of ' with 'According to'.

Reply 9: We change the  'Depending  on  the  research  to' with 'According to'. L75

Question 10: Line 57: Please add 'the' before largest important.....

Reply 10: We add 'the' before largest importan. L76

Question 11: Line 68: I think it should be 'desertified land' instead of 'esertified land'

Reply 11: We have modified it as 'desertified land'. L87

Question 12: Line 78: Please delete 'so as'

Reply 12: We delete 'so as'. L100

Question 13: Line 81: Please add the comprehensive study sites map.

Reply 13: We modified Table 1 to Figure 1 to express the comprehensive site of the study. L293

Question 14: Line 95: Table and figures should not be mixed because there is no symmetry between them. So, please separate them for easy understanding.

Reply 14: We modified Table 1 to Figure 1 to express the comprehensive site of the study. L293

Question 15: Line-103: Please consider cite/add these reference here.

https://doi.org/10.1038/s41467-021-25641-0

Reply 15: We added this reference. L523

Question 16: Line 104: Please consider cite/add these reference here.

https://doi.org/10.1038/s41467-021-25641-0

Reply 16: We added this reference. L523

Question 17: Line 162: A table/figure should be self-explanatory, so please give detail in the Table 2's caption.

Reply 17: We've combined table 1 and table 2, Add "This table shows the quantified Ecological widths of 17 major species in different stages." after the title for clarification. L123

Question 18: Line 191-192: The figure should be self-explanatory, so please give detail in the Figure 1's caption.

Reply 18: We added "Values 1-16 outside the table represent different plant species. ND stands for notBOC, LD for light desertification, MD for moderate desertification,  HD for heavy desertification. See Table 1 for the meanings of species Numbers. "to clarify. L152

Question 19: Table 4: Please add space before and after '>'.

Reply 19: We add and check to fix similar problems throughout the text. L168

Question 20: Line 213-214: The figure should be self-explanatory, so please give detail in the Figure 2's caption.

Reply 20: We added " This graph represents the significance level of the chi-square test for different plant species. ND stands for notdesertification, LD for light desertification, MD for moderate desertification, HD for heavy desertification. See Table 1 for the meanings of species numbers. "to clarify. L175

Question 21: Line 228-229:Figure is not clear so please redraw the figure clearly. Moreover,the figure should be self-explanatory, so please give detail in the Figure 3's caption.

Reply 21: We redrew the figure. L193

Question 22: Line 242-243: The figure should be self-explanatory, so please give detail in the Figure 4's.

Reply 22: We added " ND stands for notdesertification, LD for light desertification, MD for moderate desertification, HD for heavy desertification. See Table 1 for the meanings of species numbers. "to clarify. L210

Question 23: Line 260-262: Please delete

Reply 23: We delete it. L233

Question 24: Line 283: Please add space before and after '>'.

Reply 24: We've normalized the format of all the inequalities. L183-L186, L200-L204, L253

Reviewer 3 Report

Please find document attached.

Author Response

Question 1: The chapters provided for the authors are 1. Introduction; 2. Results; 3. Discussion; 4. Materials and methods; 5. Conclusions. The order in this paper is: Introduction; 1. Materials and methods; 3. Results; 4. Discussion; 5. Conclusion. Please, check the authors’ guidelines.

Reply 1: We revise the paper in the order of the author's guide.

Question 2: The References section does not include the links (doi) of the cited articles. For reasons of formality, the officially used abbreviation of the cited journals should be used. This is not currently the case.

Reply 2: We downloaded the "endnote" bibliographic reference format for the journal from the official website, added the DOI of the cited literature, and corrected the journal abbreviation. L393-L527

Question 3: Abstract: The structure of this part of the manuscript should be improved to better highlight the scientific added value of the project. Based on the Abstract, one should be able to formulate a new hypothesis. In the Abstract I propose to add numerical data, results of calculations. In line 16, the expression in this study appears twice. In line 27, the "moderate disturbance hypothesis" is not meaningful.

Reply 3:(1)We came up with " we still do not know the mechanism of how the plants were preserved and how the retained plants adapted to the new environment during the desertification process. We can further study these questions in the next step.”L29-L31(2)We added ”Stellera chromosome(3.90), Thermopsis lanceolate(3.52) and Aster almanacs(3.99) had larger niche widths”L20(3)We delet the in the study and "moderate disturbance hypothesis". L17

Question 4: Introduction: Clarifying the knowledge gaps requires a deeper explanation. Please, define the purpose of the study and its significance, and introduce the current state of the research citing the key publications. Please, highlight controversial and diverging hypotheses where relevant.

Highlighting the main up-to-date knowledge based on the published conclusions will help to clarify the objectives. At the end of this section, please, list your objectives, and reflect to them in the Results and Discussion sections strictly.

Reply 4: (1) We added the research progress on ecological niche and interspecific association, and found that there were few studies on desertified grassland in the alpine region. We hypothesized that "although the climatic and topographic factors of the Qinghai-Tibet Plateau are unique, the overall trend of vegetation competitive succession in the desertified process is basically the same". L49-L71  (2) In our discussion, we compared the studies of others, and our results are basically consistent with some of their results.

Question 5: The author of literature number 13 in line 56 does not match in text and references.

Reply 5: We confused the author's first and last name and we have corrected the error in the article. L75

Question 6: Materials and methods: I propose to insert a map of the study area. The information in this chapter should be clarified and restructured.

Reply 6: We added a map and rewrote the section. L296

Question 7: Figure 1 slids across two sheets. The abbreviations in a figure should be explained in the title of it.

Reply 7: We have added this in the title. L153

Question 8: The "Grading Index of Natural Grassland Degradation, Desertification and Salinization" (GB 19377-2003) document should be added to the reference list.

Reply 8: We have added a reference link to this data.

Question 9: The title of Table 2 (Main species and their numbers) needs to be clarified.

Reply 9: We clarify this content in the title.

Question 10: After interspecies association (line 126), how do you enter 1)Statistical magnitude (line 130) and 2)Connection strength (line 141)?

Reply 10: a is the quadrat where both species appear, b and c are the quadrat numbers where only species 1 and species 2 appear, respectively, and d is the quadrat number where neither species appear. When ad≥bc, we use the formula (10); bc>ad and d≥a, we use the formula (11); bc>ad and d>a, we use the formula (12).

Question 11: The section 1.2.2 Computational formula should be checked for the line numbers.

Reply 11: We checked the content and fixed the error. L324-L356

Question 12: The Material and methods and Results chapters should be harmonised.

Reply 12: We adjust the order of these chapters to suit the requirements of the journal.

Question13: In line 134 In formula (7) correctly in formula (8). In line 142, in formula (8)(9)(10)(11), correctly in formula (9)(10)(11) (12).

Reply 13: We rechecked and corrected the error in formula number. L324-L356

Question 14: Results: This part of the manuscript is well-detailed, properly introducing the findings. The font size of the figures could be closer to the text to make them look nicer.

Reply 14: We set the number size of the text to a small fifth so that the text is uniform.

Question 15: Table 3 has two pages.

Reply 15: We adjusted its form. L123

Question 16: In Figure 3, the abbreviation AC should be written out.

Reply 16: We added the full name "association coefficient". L193

Question 17: In Figure 4, the abbreviation OI shall be written out.

Reply 17: We added the full name "Ochiai index". 212

Question 18: Discussion: This section should be supplemented with further relevant literatures. The number of literature cited in this section (10) is modest.

Reply 18: We have added six new references. [28] [32] [33] [34] [39] [40]

Question 19: Conclusions: to make this section self-explanatory, please focus on your own findings and their explanation, reflecting each aspect in the results chapters. This should reflect one by one the objectives that should have been listed at the end of the Introduction. This should be supplemented. It should also indicate the added value of the project, with reference to the knowledge gap covered by the study. In addition, future research opportunities should be indicated. On the basis of this part of the article, you should be able to identify a new scientific problem and develop a new research plan.

Reply 19: We rewrote the conclusion. “With the intensification of desertification, the ecological niche of plateau grassland was gradually differentiated, the competition was gradually reduced, and the regional coexistence pattern of plant community was observed This study simulates the degradation succession process of main plant species niche and kind of connection between the change situation, for the study of desertification in the succession of plant community characteristics provides a new method, at the same time, to explore the desert plant competition mechanism in the environment provides a quantitative method The severe desertification in the Qinghai-tibet plateau and the world of alpine grassland recovery is very important. However, we still do not know the mechanism of how the plants were preserved and how the retained plants adapted to the new environment during the desertification process. We can further study these questions in the next step.” L365-L375

Reviewer 4 Report

This paper presented some analysis on the species interrelation along a degradation gradient in the Qinghai lake region. The questions, the data are  good.

The paper is interesting. My main comment is regarding the presentation of the method and the results.

The authors use several indexes based on the botanical surveys. They explain the calculation but I think the ecology behind each index could be more presented especially in the method (“we used this index to evaluated the interaction between species”).

And they make the calculation of each one of their 17 species and in the four situations. This produced a lot of number.

I am not sure of the graphical representation especially the Figure 1, figure 2, Figure 3 and Figure 4. I think the important question is the change of the indexes between the level of degradation. I will suggest to make plot with on the X axis the value on one situation of degradation and the y axis the value on another situation of degradation. Each point will a species or a pair of species. This way it will be easier to compare the situation than with actual figure.

If The authors want to presented the interaction between species I, they could used graphical representation adapted to distance (dendrogram for example).

More, I think the text of the results contains to much number et It will be better to summarise the results otherwise  the reader is a bit lost.

I have few small comments

I will prefer a map of the different sites than a table with the coordinates ( I don’t understand really the picture). With site theses correspond

I think it is better to use abbreviation than number of the species . It is easier to follow for the people that know the species.

The numerotation of the equation are  different between the text and the equation  for the equation(8).

Author Response

Question 1: The authors use several indexes based on the botanical surveys. They explain the calculation but I think the ecology behind each index could be more presented especially in the method (“we used this index to evaluated the interaction between species”).

Reply 1 : In the method section, we added the ecological significance represented by the index at the appropriate position. L313, L340

Question 2: I am not sure of the graphical representation especially the Figure 1, figure 2, Figure 3 and Figure 4. I think the important question is the change of the indexes between the level of degradation. I will suggest to make plot with on the X axis the value on one situation of degradation and the y axis the value on another situation of degradation. Each point will a species or a pair of species. This way it will be easier to compare the situation than with actual figure.

Reply 2: This is an excellent idea. However, we tried to illustrate the interspecific association and the overall association of species. The matrix diagram we used showed the numerical changes and also compared the near and far relationship between multiple species. We thought that the two-dimensional coordinate system could not reflect this point, so this suggestion was not used.

Question 3: More, I think the text of the results contains to much number et It will be better to summarise the results otherwise  the reader is a bit lost.

Reply 3: We reviewed and revised the content of the full text, which is formatted as "summary + data exposition" or "data exposition + summary ".

Question 4: I will prefer a map of the different sites than a table with the coordinates ( I don’t understand really the picture). With site theses correspond

Reply 4: We have replaced the original standard 1 with Figure 1 (location map and sample point annotation) to accommodate this suggestion.

Question 5: I think it is better to use abbreviation than number of the species . It is easier to follow for the people that know the species.

Reply 5: We tried to use the abbreviation of the plant species instead of the number, but unfortunately, with the abbreviation of the plant, our picture was out of place, so we had to still use the number instead of the species name.

Question 6: The numerotation of the equation are  different between the text and the equation  for the equation(8).

Reply 6: We are very sorry for this error. We have corrected it and checked the numbering and text of all formulas. L324-L356

Round 2

Reviewer 1 Report

The quality of the manuscript has been significantly improved by the authors' revisions. The revised manuscript is recommended for publication in this journal.

Author Response

Dear reviewer, thanks for your selfless dedication to this manuscript. I think it is because of your work that researchers are able to publish their research results quickly. We have never met, but your contribution to the scientific cause of the world is worthy of praise.

Reviewer 3 Report

The authors have taken my suggestions into account in the revised manuscript. The quality of the manuscript has been significantly improved. I recommend publication of the manuscript in the journal.

Author Response

Dear reviewer, thanks you for your selfless dedication to this manuscript. I think it is because of your work that researchers are able to publish their research results quickly. We have never met, but your contribution to the scientific cause of the world is worthy of praise.

Reviewer 4 Report

Dear Authors

I am still not convince to the type of graphique used and the clearity of this graphique but  it is the editor choise.

I have also the impression that some of the editing is strange ( Material and methods at the end and so on)

otherwise , the paper is clear

All the best

Author Response

Dear reviewer, thank you for your selfless dedication to this manuscript. I think it is because of your work that researchers are able to publish their research results quickly. We have never met, but your contribution to the scientific cause of the world is worthy of praise.

Our reply to your question is as follows,

Question:

  • I am still not convince to the type of graphique used and the clearity of this graphique but  it is the editor choise.
  • I have also the impression that some of the editing is strange ( Material and methods at the end and so on)

Answer: This is an excellent question, however, in order to follow the publication format of the journal, we regret that we were not able to revise the manuscript as you suggested.
